# Monitoring New Long-Lasting Intravitreal Formulation for Glaucoma with Vitreous Images Using Optical Coherence Tomography

**DOI:** 10.3390/pharmaceutics13020217

**Published:** 2021-02-05

**Authors:** Maria Jesus Rodrigo, Amaya Pérez del Palomar, Alberto Montolío, Silvia Mendez-Martinez, Manuel Subias, Maria Jose Cardiel, Teresa Martinez-Rincon, José Cegoñino, José Maria Fraile, Eugenio Vispe, José Antonio Mayoral, Vicente Polo, Elena Garcia-Martin

**Affiliations:** 1Department of Ophthalmology, Miguel Servet University Hospital, 50009 Zaragoza, Spain; silviamendezmartinez@hotmail.com (S.M.-M.); manusubias@gmail.com (M.S.); teresamrincon@gmail.com (T.M.-R.); vpolo@unizar.es (V.P.); egmvivax@yahoo.com (E.G.-M.); 2Miguel Servet Ophthalmology Research Group (GIMSO), Aragon Health Research Institute (IIS Aragon), University of Zaragoza, 50009 Zaragoza, Spain; 3RETICS: Thematic Networks for Co-Operative Research in Health for Ocular Diseases, 28040 Madrid, Spain; 4Biomaterials Group, Aragon Institute of Engineering Research (I3A), University of Zaragoza, 50018 Zaragoza, Spain; amaya@unizar.es (A.P.d.P.); amontolio@unizar.es (A.M.); jcegoni@unizar.es (J.C.); 5Department of Mechanical Engineering, University of Zaragoza, 50018 Zaragoza, Spain; 6Department of Pathology, Lozano Blesa University Hospital, 50009 Zaragoza, Spain; mjcardielgarcia@gmail.com; 7Institute for Chemical Synthesis and Homogeneous Catalysis (ISQCH), Faculty of Sciences, University of Zaragoza-CSIC, C/Pedro Cerbuna 12, 50009 Zaragoza, Spain; jmfraile@unizar.es (J.M.F.); mayoral@unizar.es (J.A.M.); 8Chromatography and Spectroscopy Laboratory, Institute for Chemical Synthesis and Homogeneous Catalysis (ISQCH), Faculty of Sciences, University of Zaragoza-CSIC, Pedro Cerbuna 12, 50009 Zaragoza, Spain; evp@unizar.es

**Keywords:** brimonidine, Laponite, drug delivery, glaucoma, nanomedicine, monitoring, optical coherence tomography, vitreous, intravitreal

## Abstract

Intravitreal injection is the gold standard therapeutic option for posterior segment pathologies, and long-lasting release is necessary to avoid reinjections. There is no effective intravitreal treatment for glaucoma or other optic neuropathies in daily practice, nor is there a non-invasive method to monitor drug levels in the vitreous. Here we show that a glaucoma treatment combining a hypotensive and neuroprotective intravitreal formulation (IF) of brimonidine–Laponite (BRI/LAP) can be monitored non-invasively using vitreoretinal interface imaging captured with optical coherence tomography (OCT) over 24 weeks of follow-up. Qualitative and quantitative characterisation was achieved by analysing the changes in vitreous (VIT) signal intensity, expressed as a ratio of retinal pigment epithelium (RPE) intensity. Vitreous hyperreflective aggregates mixed in the vitreous and tended to settle on the retinal surface. Relative intensity and aggregate size progressively decreased over 24 weeks in treated rat eyes as the BRI/LAP IF degraded. VIT/RPE relative intensity and total aggregate area correlated with brimonidine levels measured in the eye. The OCT-derived VIT/RPE relative intensity may be a useful and objective marker for non-invasive monitoring of BRI/LAP IF.

## 1. Introduction

Brimonidine is an ocular hypotension drug widely used in ophthalmological clinical care. For adequate therapeutic control, topical administration is recommended twice a day because of its short half-life (approximately 12 h) [1]. Recently, brimonidine also exhibited a neuroprotective effect in retinal ganglion cells (RGCs), photoreceptors and other retinal cells [2] in animal and human studies [3] and with both topical and intravitreal administration [4]. The short half-life of brimonidine means that periodic intravitreal administration is needed to achieve therapeutic efficacy. However, repeated injection could lead to complications [5]. Therefore, development of sustained drug delivery systems is necessary. Laponite^®^ Na^+0.7^[(Si_8_Mg_5.5_Li_0.3_)O_20_(OH)_4_]^−0.7^ is a biocompatible and biodegradable clay used in biomedicine [6]. Its safety in intravitreal injection has been demonstrated in animal studies [7,8,9]. Laponite^®^ forms a transparent gel when dispersed in an aqueous medium. It is able to interact with other molecules, acting as a carrier and releasing the drug in a controlled manner [10]. In this regard, our research group recently demonstrated that an intravitreal formulation (IF) containing brimonidine–Laponite (BRI/LAP) produced a hypotensive and neuroprotective effect in a chronic glaucoma animal model over 6 months of follow-up [9].

The vitreous humour is a 3-D structure mainly composed of water, collagen fibres and hyaluronan that occupies 80% of eye volume. It allows light transmission and intervenes in eye metabolism [11]. A vitreous sample biopsy is a useful means of diagnosing inflammatory, infectious or oncological ocular diseases as well as of carrying out toxicological post-mortem analysis [12,13]. Furthermore, the vitreous humour is becoming increasingly important in clinical practice as it acts as a therapeutic target for posterior pole pathologies involving macular oedema, such as diabetic retinopathy, vascular occlusions or aged-related macular degeneration. However, as an intravitreal treatment for glaucoma is not yet available in clinical practice, many researchers are working to resolve this limitation within a medium-term horizon. The potential interest in using intravitreal administration is to circumvent ocular barriers while avoiding systemic adverse events and, by acting as a reservoir, maintaining therapeutic drug levels near the site of action [14].

Optical coherence tomography (OCT) is an easy-to-handle, cost-efficient, and objective technology that provides high-resolution cross-sectional images. It has long been widely used in clinical practice and research to study neuroretinal structure. Improved OCT devices with swept-source or enhanced vitreous imaging have made it possible to study the vitreous in normal and inflamed states and to study changes to it after treatment [15,16,17]. However, it has never been used to monitor drug levels in IFs. Currently, loss in IF therapeutic efficacy is evaluated on the basis of changes in neuroretinal structure or decreases in subjective visual acuity measurements, which indicate an increase in disease activity. It would be beneficial, however, to find an objective vitreous monitoring marker with which to quantify the degradation of the injected molecule and which could even anticipate the loss of therapeutic effect before structural retinal changes are detected by OCT, especially as compared with other more expensive or invasive techniques used in research, such as positron emission tomography or magnetic resonance imaging [18,19].

In previous OCT-based studies by this group, BRI/LAP IF was visualised in the vitreous cavity as hyperreflective aggregates [9]. Expanding on this finding, this study uses OCT to analyse changes in the hyperreflectivity signal from the BRI/LAP IF in the vitreous humour in rat eyes over a 24-week period. This paper describes a monitoring marker for the IF, correlates the OCT vitreous signal to drug levels and discusses the therapeutic effect of BRI/LAP IF [9]. It proposes serial OCT as a more affordable and simpler method for monitoring BRI/LAP IF in vivo in animal research.

## 2. Materials and Methods

### 2.1. Data Collection

OCT images of the vitreoretinal interface and drug level data were obtained from the experiments carried out in a previous interventional study conducted by the authors [9] (CC BY 4.0 license). In that study, chronic glaucoma was bilaterally induced by biweekly injections of hypertonic solution into the episcleral veins, according to the well-established Morrison model [20], which led to episcleral vein sclerosis and therefore ocular hypertension (OHT). Thereafter, a single 3-µL BRI/LAP IF injection (10 mg BRI/LAP/mL) into the vitreous cavity of right eye (RE) of Long–Evans rats was performed at baseline. Left eyes (LEs) served as non-treated hypertensive controls. Intraocular pressure was measured with a rebound tonometer (Tonolab^®^ iCare, Helsinki, Finland) for rodent research. The material and methods used for the study were deeply detailed in [9]. In this study, OCT examinations of the 43 treated rats at baseline and at weeks 1, 2, 4, 6, 8, 12, 24 after intravitreal injection were analysed to quantify BRI/LAP IF evolution over 6 months. Another 23 non-treated healthy rats were examined for comparison as non-hypertensive controls. The experiment was approved beforehand by the Ethics Committee for Animal Research of Zaragoza University (PI34/17, 27th June 2017) and was carried out in strict accordance with the Association for Research in Vision and Ophthalmology’s Statement for the Use of Animals.

### 2.2. Optical Coherence Tomography

Images were acquired using a high-resolution OCT device (HR-OCT Spectralis, Heidelberg^®^ Engineering, Heidelberg, Germany) with a plane power polymethylmethacrylate (PMMA) contact lens of 270 μm thickness and 5.2 mm diameter (Cantor+Nissel^®^, Northamptonshire, UK) adapted to the rats’ cornea to obtain higher quality images. The rodent version of this system acquires cross-sectional images by means of 61 b-scans measuring around 3 mm in length and centred on the optic nerve. It has a resolution of 3 microns per pixel generated. A total of 1536 × 496 pixels per image were analysed. The retinal posterior pole protocol with automatic segmentation, eye-tracking software and follow-up application were used to ensure that the same points were re-scanned throughout the study. The “enhance depth imaging” mode was disabled in all cases.

### 2.3. Brimonidine–Laponite (BRI/LAP) Formulation and Analysis

BRI/LAP was prepared [9] by addition of Laponite (100 mg) to a solution of Brimonidine (10 mg) in ethanol (10 mL) with stirring and then solvent evaporation under a vacuum, to obtain the BRI/LAP formulation in powder form, which was gamma-ray sterilised. BRI/LAP was injected in the form of a yellow colloidal dispersion in balanced saline solution (BSS) (10 mg/mL).

The brimonidine content in the rat eyes was analysed [8,9] using an ultra-high-pressure liquid chromatography mass spectrometer (UHPLC-MS, Waters, Milford, MA, USA). The eyes were first cut into pieces, sonicated with a solution of formic acid in acetonitrile, then with ammonium formate in phosphoric acid and internal standard (2-bromoquinoxaline), centrifuged and the supernatant was cleaned up by solid phase extraction.

### 2.4. In Vitro Release of BRI from BRI/LAP Formulation

Release was studied in a model of vitreous humour made up of 0.5% sodium hyaluronate in saline solution (pH 7.1–7.4). The release tests were performed by dispersing BRI/LAP (5 mg, weight ratio 1/10, brimonidine amount 454.5 μg) in the extraction medium (0.5 g of the vitreous model) under stirring at 120 rpm at 37 °C. After 24 h the dispersion was centrifuged at 14,000 rpm for 20 min. The liquid phase was separated and the solid was re-dispersed in fresh extraction medium for a new cycle. The liquid phase was diluted with acetonitrile (0.5 mL) containing the internal standard, centrifuged again and analysed by HPLC as described above.

### 2.5. Analysis of the Intravitreal Formulation Using OCT

The BRI/LAP IF aggregates present in the OCT scans of the vitreoretinal interface were studied. This technique focuses on the analysis of opacities in the vitreoretinal interface by OCT, which does not require a correction factor for its histological correlation [21] and ensures a characterisation of the actual aggregate. These aggregates were defined as being dots dispersed in the vitreous humour or vitreoretinal interface whose larger size, irregularity or greater signal hyperreflectivity differentiated them from background speckle noise.

OCT image contrast was not adjusted at any time. OCT raw images were exported as Audio Video Interleave (AVI) videos. These videos were analysed using a custom program implemented in Matlab (version R218a, Mathworks Inc., Natick, MA, USA). This code allows us to find the inner limiting membrane (ILM) and the inner and outer layers of the retinal pigment epithelium (RPE) by grayscale conversion. This makes it possible to delimit vitreous space and RPE space in each b-scan (Figure 1). The vitreous was defined as the space between the uppermost extent of the b-scan and the ILM, whereas the RPE was defined as the space between the inner and outer layer of the RPE [16,17]. The mean intensity value of these two spaces was calculated as the average of the intensity of all the pixels within each region, obtaining the VIT/RPE relative intensity in each b-scan. Thus, the VIT/RPE relative intensity of each eye is the average of the 61 b-scans. Furthermore, as the BRI/LAP aggregates tend to be deposited on the ILM–retinal nerve fibre layer (RNFL), the inner and outer limits of the ILM–RNFL were determined to obtain the evolution of ILM–RNFL thickness throughout the 24 weeks of follow-up.

The size of the aggregates in each b-scan was also determined by calculating the number of pixels that each aggregate contains in the image. In the analysed b-scans, there are a total of 761,856 pixels and the image area is 2906 mm^2^. Therefore, the ratio is 3815 µm^2^/pixel. To calculate that correctly, the background speckle noise of the image was deleted using a denoising filter, which made it possible to distinguish between aggregates and background noise. This filter was implemented in our custom code following the definition of aggregates as larger dots whose intensity is greater than the background intensity. In order to ensure that we only quantify the aggregates produced by the BRI/LAP IF, a minimum limit was established, so that the possible noise [22] due to the physiological components of the eye were not taken into account. This minimum limit was set at 500 µm^2^ per aggregate. Once we had calculated the size of each aggregate, we could compute the average area and total area of the BRI/LAP IF in each eye at different stages during the follow-up.

The imaging data were analysed for clinical and drug level information by a masked reader. OCT segmentation was performed by two different researchers, likewise masked, to verify reproducibility.

### 2.6. Statistical Analysis

All data were recorded in an Excel database, and statistical analysis was performed using SPSS software version 20.0 (SPSS Inc., Chicago, IL, USA). To assess sample distribution, the Kolmogorov–Smirnov test was used. However, given the non-parametric distribution of most of the data, the Mann–Whitney U test was employed to evaluate the differences between both cohorts, and a paired Wilcoxon test was used to compare the changes recorded in each eye over the study period. *p* values < 0.05 were considered to indicate statistical significance.

## 3. Results

A total of 186 OCT videos from 43 treated rats (43 hypertensive REs treated with BRI/LAP IF and 29 non-treated hypertensive LEs) and 23 non-treated healthy rats (23 REs/23 LEs) were analysed. The REs injected with the BRI/LAP IF showed hyperreflective dots/aggregates mixing uniformly in the vitreous gel and dispersed as floaters, with a tendency to move toward the vitreoretinal interface during the 24-week follow-up [9]. There was also OCT-guided evidence of the hyperreflective dots crossing the vitreoretinal interface and embedding deeply in the retinal tissue. Particular qualitative characteristics of the behaviour of the BRI/LAP IF observed in several animals are shown in Figure 2.

OCT also detected a progressive decrease over time in the number and size of BRI/LAP aggregates (Figure 3). Figure 4 shows the temporal change in aggregate size. Figure 4A shows that total aggregate area decreased with time over the 24 weeks of follow-up. This drop was very marked during the first 4–6 weeks, after which the change was more gradual. Moreover, total aggregate area increased two weeks after injection. Although a similar trend was observed (aggregate size increased at 2 weeks), this seems to be at the expense of an increase in mean aggregate size. A considerable decrease was then detected around 8 weeks, and from 12 weeks onwards mean aggregate size remained practically constant (Figure 4B).

OCT analysis revealed the decreasing intensity of the hyperreflective IF aggregates in the vitreous over 24 weeks of follow-up (Figure 5A). The figure shows that the intensity peaked at the end of the second week and then decreased until it stabilised around week 12. It should be noted that the intensity index rose slightly from weeks 4 to 6 and that this coincided with a decrease in total aggregate area (Figure 4A) and a slight increase in mean aggregate area (Figure 4B). Comparison of the intensity indices for REs (treated with BRI/LAP IF) and LEs (non-treated) (Figure 5A) shows that the intensity is very much lower in the LEs (0.30 vs., 0.25; *p* < 0.001). As can be seen, VIT/RPE relative intensity in eyes with glaucoma treated with BRI/LAP IF is higher than in eyes with non-treated glaucoma due to the presence of BRI/LAP IF. Interestingly, the index similarly increased over the first two weeks. This effect could be produced by the induction of glaucoma. Furthermore, the intensity index for non-treated eyes (right and left control eyes; the grey lines in Figure 5A) was also computed. In this case, the intensity value remained constant (0.17). Here, the difference observed in the intensity index between eyes with glaucoma and healthy controls is produced by glaucoma induction. Finally, we observed that the aggregates were initially distributed throughout the vitreous but as time went by (6 to 8 weeks) they settled on top of the ILM–RNFL. It was not possible to distinguish between the aggregates and the ILM–RNFL because the intensity values are very similar. Therefore, this co-layer was segmented in order to measure the deposited aggregates. Figure 5B shows how RE ILM–RNFL thickness increased until it plateaued at week 12. This increase in thickness is directly related to the aggregates’ distribution on top of the retina. In the same plot, LE (untreated) ILM–RNFL thickness remained unaltered or decreased slightly during the follow-up if no aggregates were present. Furthermore, we previously not only ruled out that the thickness increase in the treated eye was a consequence of neurodegeneration or cystoid oedema, but also observed functional neuroprotection and a higher RGC count [9].

Finally, a 3-D reconstruction of the 61 b-scans from a specific rat was performed in order to assess the qualitative decrease in aggregates with time. Figure 6 shows the same eye at 2 weeks of follow-up and then 6 weeks later (8 weeks of follow-up). It clearly shows that the aggregates are widely dispersed 2 weeks post-injection, and that 6 weeks later the aggregates are fewer and smaller and have practically disappeared from the vitreous humour.

In order to investigate if the vitreous OCT data could serve as an objective marker for non-invasive monitoring of the IF, the curve of the brimonidine levels extracted from our previous study [9] (CC BY 4.0 license) was correlated with the VIT/RPE relative intensities and with the curve of the total aggregate area (as an expression of the total amount of IF injected) obtained using OCT at weeks 1, 4, 8, and 24 after intravitreal injection. Both the brimonidine levels and the VIT/RPE relative intensity curves showed a negative linear tendency with a direct correlation (y = −0.0003x + 0.1016 R^2^ = 0.5616 vs. y = −0.0002x + 0.2543 R^2^ = 0.4301, respectively). Moreover, the logarithmic curves of the brimonidine levels and the total aggregate area were very similar (Figure 7).

A short in vitro study was performed to compare with the results obtained in the in vivo study. A model for vitreous humour (VHM) formed by sodium hialuronate in saline solution was chosen as medium for release, and the procedure was analogous to our precedent study with the DEX/LAP system [23], with equilibration of the BRI/LAP formulation in the VHM for 24 h, centrifugation to separate the liquid phase with the released BRI for analysis and re-suspension of the solid in a new batch of VHM. As can be seen in Figure 8, the released amount was higher in the first extractions, indicating the presence of a fraction of BRI loosely bound to LAP, whereas the released amount in the successive extractions is much lower, corresponding to the BRI fraction more tightly bound to LAP. In any case, the total amount released after eight extractions is lower than the 22% of the total BRI present in BRI/LAP, confirming in this way the ability of this formulation for a sustained release for a long time period.

## 4. Discussion

This paper describes use of OCT to perform non-invasive monitoring of an intravitreal formulation (BRI/LAP) used to treat glaucoma.

Qualitative study made it possible to observe the behaviour of BRI/LAP IF in the vitreous and retina over a 24-week period [9]. In the early stages of the study the BRI/LAP IF was mixed in the vitreous, remaining in suspension in small microaggregates that later showed a tendency to approach and attach to the retina, possibly due to brimonidine tropism towards the alpha-adrenergic receptors present in the ganglion cell layer, inner nuclear layer and outer nuclear layer [24], melanin, the RPE and the choroid. In animal studies, brimonidine has been shown to have both a functional and a structural neuroprotective effect. Intravitreal administration of brimonidine-loaded nanoparticles has shown a neuroprotective effect over 14 days of monitoring [4], a hypotensive and neuroprotective effect lasting 4 weeks in an acute glaucoma model [25] and, recently, our group [9] demonstrated a functional and structural hypotensive and neuroprotective effect lasting 6 months with BRI/LAP IF in a chronic glaucoma model. This was proved not only by analysing the neuroretinal thickness with OCT and the functionally with electroretinography, but also by analysing images of the vitreoretinal interface obtained with OCT, which is a novel measurement method that could provide a non-invasive, objective and reliable means of monitoring the pharmacodynamics of the IF. In addition, brimonidine has been shown to be effective in clinical trials with human patients when administered topically to treat diabetic retinopathy [3] and intravitreally with a brimonidine drug delivery system (Brimo DDS Allergan^®^, Irvine, CA, USA) in geographic atrophy from age-related macular degeneration [26]. The administration of Brimo DDS^®^ requires an applicator and therefore larger gauge needles, as well as re-implantation every 3 months. A potential sustained-release intravitreal injection-based therapy such as BRI/LAP IF could be effective and useful in these patients as it offers advantages such as injection with smaller micrometric needles, nanoscale formulation and longer therapeutic effects lasting up to 6 months [9].

Repeated intravitreal injections can increase intraocular pressure (IOP) and alter the structure of the optic nerve [27], which can hinder or skew diagnosis and follow-up based on OCT measurements, which are very important in assessing progression in glaucoma. Hence, the use of sustained release systems is necessary in order to reduce re-injection and risk to patients. In this regard, Laponite produced sustained release of drugs for at least 6 months [7,8,9] and improved the solubility of hydrophobic substances such as brimonidine, thereby facilitating their administration and diffusion within the vitreous.

For the formulation injected into the vitreous to reach the retinal cells (and exert its effect), it must pass through the posterior vitreous cortex (PVC) as well as the ILM, which has pores measuring 10–20 nm. OCT showed hyperreflective aggregates capable of traversing the PVC and ILM and reaching the intraretinal space, possibly by diffusion (Figure 2) due to the small size of the Laponite platelets (1 nm high, 30 nm diameter) [6] and the lipophilicity of brimonidine. In this regard, various authors also found internalisation of drug delivery systems in the retina. Koo et al. [28] described the passage of intravitreal nanoparticles through the retina by both diffusion and endocytosis by the Müller cells, and Xu et al. [29] found that cationic amphiphilic intravitreal polymers reached RPE cells. The space between the ILM and the PVC houses an interdigitate extracellular matrix [15]. In this space, the authors observed that the BRI/LAP IF arranged itself in a row with the aggregates ordered one after the other at different heights, concentrically (see Figure 2A,C and Figure 6). This arrangement is similar to that proposed in the formation of Laponite film, [30] exhibiting a side-by-side arrangement of the Laponite platelets and subsequent stacking of them in different layers. This arrangement is favoured by several film-forming methods, including the Langmuir–Blodgett method, in a liquid–solid interface similar to the vitreous–retina interface, and even more so in the case of hybrid films with organic molecules and macromolecules [31].

OCT was fundamental to demonstrate in vivo degradation of the amount of BRI/LAP IF injected (based on total aggregate area) and morphological and dynamic aggregation of the Laponite molecules (based on mean aggregate area). Acidosis [32] and inflammatory proteins [33] have been detected in the vitreous of glaucoma patients. In both situations, Laponite degradation increases. In our in vivo study (coinciding with the in vitro study), the highest rate of BRI/LAP IF degradation with brimonidine release occurred in the early stages (the largest decrease in total aggregate area occurred in the first 2–6 weeks (Figure 4A)). However, between the middle of the study and the end (at 24 weeks) degradation happened very slowly and the size of the aggregates remained stable. In our previous paper [9], we demonstrated that the BRI/LAP IF produced an early hypotensive effect until 6–8 weeks (peaking at 2 weeks) (see Table 1) that coincided with higher levels of brimonidine in the eye, with the greatest aggregates’ area (Figure 4B) and relative intensities of VIT/RPE (Figure 5A), as well as a neuroprotective effect mainly occurring in the later stages (with sustainedly low levels of brimonidine), quantified in the form of reduced retinal RGC death. This reduced RGC death, which brimonidine achieves by blocking the excitotoxicity of the glutamate [34], is probably why the vitreous acidosis was maintained or did not increase and, consequently, why the BRI/LAP IF degradation rate was slower in the later stages. In addition, Laponite aggregates in situations in which pH is very low (acidosis) [35]. Greatest aggregation observed using OCT (measured as an increase in the area of the aggregate) was found in the early stages, coinciding with the onset of the damage induced by ocular hypertension and, therefore, increased acidosis. However, at later stages the aggregates were much smaller, suggesting less acidosis. Both OCT analyses of the aggregates (degradation based on total aggregate area, and aggregation based on mean aggregate area) support the idea of lower cell death due to neuroprotection in the later stages. This suggests a possible correlation/association between analysis of the vitreous using OCT with BRI/LAP IF and the results of structural analysis using OCT and retinal histology [9].

The OCT study of the vitreoretinal interface was also helpful in understanding a phenomenon that was unresponsive in our previous study [9]. We found that although BRI/LAP IF generally exerted a hypotensive effect on the eye treated, a peak in IOP was detected in week 3 compared to the non-treated LE. Laponite swells in aqueous media, so we hypothesised that an increase in the size of the aggregate was responsible for the IOP increase. In this study based on OCT monitoring of aggregates, as expected, an increase in size of the BRI/LAP IF was measured in the early stages (2–3 weeks), confirming our hypothesis.

Studying pharmacokinetics or tracing in relation to an IF in the visual pathway is not simple [14] as it requires invasive biopsies or tests that are either very expensive or only accessible to researchers [18]. OCT is a non-invasive technology capable of offering almost histological images depending on the light transmitted or reflected as light passes through different structures with differing densities. Those areas that light finds it harder to pass through have higher optical densities (e.g., lipid or calcium membranes), and different optical density ratios have been found according to pathology [36]. The increase in signal offered by BRI/LAP IF may be a consequence of the lipophilicity of the brimonidine and the silicon and magnesium components of the Laponite.

As OCT is a light-scattering imaging technique, higher VIT/RPE relative intensity would indicate greater light scattering, with risk of perception of floaters. In this regard, the size of the aggregates was in the order of microns (from 850 to 760 microns) (Figure 4B), much smaller than the commercially available Ozurdex^®^ implant measuring 0.46 × 6 mm (22G needle) or Brimo DDS^®^ (25G needle) [26]. Larger aggregates potentially perceivable as floaters (Figure 2D) were nevertheless observed occasionally. It was only in the largest and least frequently occurring sizes that a posterior shadow was detected. In most aggregates no shadow was detected, suggesting that potential floater perception would be minimal or non-existent. This was reinforced by the fact that the animals did not exhibit any abnormal behaviour.

Two previous studies involving intravitreal administration of Laponite and using indirect ophthalmoscopy [7,8] describe the IF as a single floccule/lump floating in the vitreous. However, in this study observations of large floccules were incidental/isolated. Laponite has thixotropic characteristics that facilitate IF injection fluidity by using smaller-gauge needles (such as the Hamilton µLSyringes^®^ used in this study) [9]. Likewise, the injection force or speed and the needle gauge used have been shown to influence turbulence and mixing after injection of aqueous and viscous solutions into the vitreous gel [37]. This study suggests that BRI/LAP IF administration using smaller-gauge needles would, in addition to causing less patient discomfort, prevent the formation of large floccules.

In this paper, OCT analysis of the vitreous shows that VIT/RPE relative intensity is (1) significantly higher in eyes treated with IF than in non-treated eyes; (2) that intensity values decrease over time; (3) that scores can be calculated with a high degree of reproducibility; and (4) that the total aggregate area correlates with the amount of brimonidine at all stages of the study and shows a similar degradation curve. Therefore, VIT/RPE relative intensity would be an objective marker. The advantage of this OCT-guided analysis of IF persistence in the vitreous would be the possibility of acquiring simple measurements in repeatable explorations at any stage of disease development. This would help detect or predict a loss of treatment efficacy, avoiding detection of the absence of therapeutic effect only after retinal structural damage occurs, in other words as the disease progresses, as is currently the case with glaucoma. Moreover, it also makes it possible to evaluate the rate of IF degradation individually per patient, thereby making precision medicine more personalised [38]. In addition, guided OCT evaluation can reduce sample size as well as cost both in animal studies and in future clinical trials with patients.

VIT/RPE relative intensity has been used because in previous studies it proved, as a marker of inflammation, to be a repeatable measure with a high degree of reproducibility and sensitivity [16]. Furthermore, Sreekantam et al. [17] mentioned that the RPE signal may be slightly attenuated in the case of macular oedema, resulting in lower VIT/RPE relative intensity. However, glaucoma does not present with oedema, so presumably the utility would be maintained in this pathology, perhaps with even greater reliability.

### Limitations of the Study

The depth of the vitreous analysed using OCT is partial and limited to a maximum of 1.9 mm. The authors consider that the analysis performed on the vitreoretinal interface using a follow-up protocol (which involves studying the exact same location in serial scans) presents a representative sample of the complete vitreous gel. This view is shared by other researchers who demonstrated the use of vitreoretinal interface analysis as a non-invasive measure correlated with eye inflammation in animal and human studies [21,39]. The high correlation shown in the results (OCT signal intensity vs. brimonidine concentration in the eye) demonstrates applicability in rats and, presented as a logarithmic scale, suggests that study of the whole vitreous body would not be necessary. However, these animals have a huge lens and therefore a very small vitreous volume compared to humans. In further studies full analysis of the vitreous body would be recommended to corroborate the applicability of this monitoring in animals with a similar organisational structure to humans, such as pigs or dogs. For that purpose, swept-source OCT would help to enhance visualisation of the vitreous cavity. Moreover, as noted above, the increased signal obtained with IF may be a consequence of lipolicity of brimonidine inserted between the silica and magnesium components of the Laponite. This technique seems to be suitable for hyper-reflective IF, in contrast to previous therapies injected into the vitreous humour. In this study, blank Laponite was not used because it was previously shown to have an intraocular durability of up to 6 months [7,8,9]. We are now focusing on IF monitoring for therapeutic application of glaucoma over a long period of time. However, it would be desirable to conduct further studies with only Laponite evaluated with OCT. This study was conducted without enhanced vitreous imaging technology, which could have helped reveal more subtle diffusion characteristics that were not found with the technology used. This research was performed using a commercially available OCT device employed in animal study and a protocol customised specifically for OCT image analysis. Automated analysis of the type that already exists for the study of the neuroretina and that would allow for graduation of signal intensity loss or its rate at the time of acquisition would be beneficial and would facilitate real-time clinical decision-making.

Another limitation could be that some of the hyperreflective dots were due to latent inflammation [16] secondary to the induction of glaucoma and alteration or increase in signal intensity. Eye infection and severe vitritis were discarded in rat eyes injected with BRI/LAP IF in our previous study [9], (CC BY 4.0 license). In this regard, the vitreous cells, hyalocytes, can be exacerbated in inflamed eyes [40] and vitreous changes are early markers of retinal damage [41]. Several studies have demonstrated the capability of monitoring acute and obvious ocular inflammation in vitreous by OCT and correlated it with retinal disease progression [16,17,21,39,42]. However, this first study opens a window to the possibility of also evaluating subclinical inflammation by vitreous imaging. To our knowledge, this is the first OCT-based study to detect an alteration in the vitreous of eyes with glaucoma (higher signal intensity) when compared with healthy eyes. As both eyes were injected to induce OHT, the increased signal intensity found in the RE is considered a consequence of BRI/LAP IF.

Considering translation to clinical settings, one issue to take into consideration would be the vitreous opacities that are normally present in older humans, which are the primary target population for this formulation. However, in general, floaters are usually kept in a stable range, and the monitoring method presented in this study is conducted with a relative signal (VIT/RPE). Therefore, the change of signal in the successive re-scans (monitoring) would be the consequence of a change in the formulation. Furthermore, in the case of pathology, the metabolism of the drugs is altered. Recently, an increase in vitreous signal VIT/RPE has been described in pathologies such as diabetic retinopathy [39], meaning future studies would be necessary to shed light on these doubts. Nevertheless, this study presents a non-invasive, cost-efficient and customisable monitoring method that would facilitate precision medicine [38].

## 5. Conclusions

This longitudinal study describes for the first time a qualitative and quantitative method of using OCT to analyse the signal generated in the vitreous by the BRI/LAP IF for glaucoma treatment. It enables initial monitoring of a therapeutic intravitreal formulation based on objective measurement of changes in the vitreous using OCT and demonstrates an adequate correlation with brimonidine drug levels. These results are a preliminary step in the validation of this potential biomarker identified with OCT for use in therapeutic monitoring, IF monitoring or tracing and could also be useful for conducting more accurate clinical trials based on early critical points of loss of efficacy. They could also potentially be transferred to clinical settings employing new OCT devices that offer higher image resolution.

## 6. Patents

J.M.F., J.A.M., V.P. and E.G.M. are inventors on a pending European patent application (No. 20 382 021.2) related to this technology. The terms of this arrangement are being managed by the Aragon Health Research Institute (IIS Aragon) and Zaragoza University in accordance with its conflict of interest policies.

## Figures and Tables

**Figure 1 pharmaceutics-13-00217-f001:**
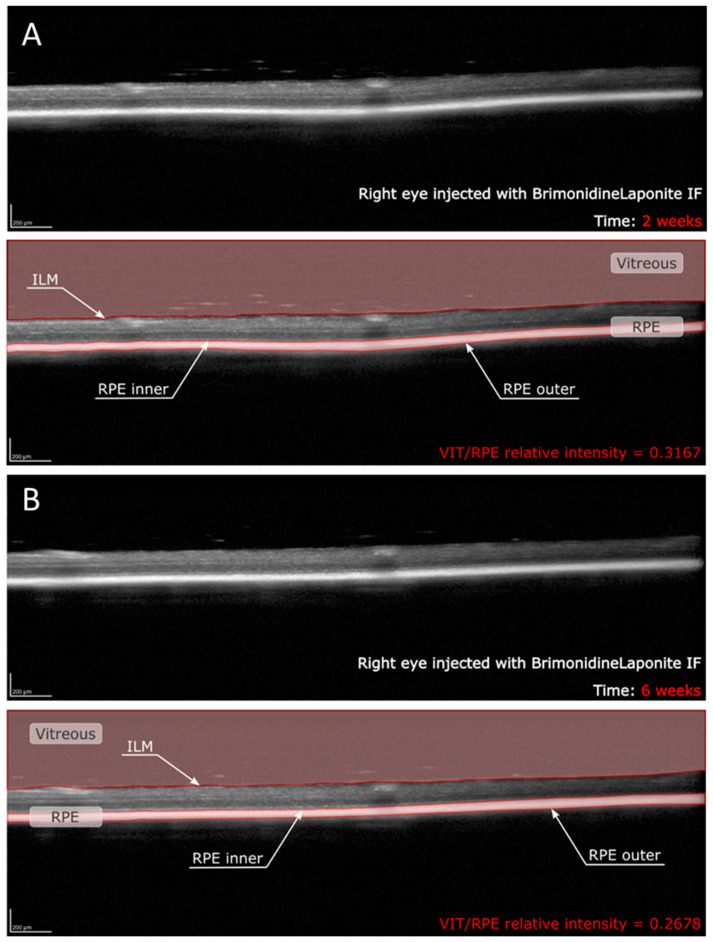
Quantitative assessment of VIT/RPE relative intensity in the same right eye of a rat at (**A**) two weeks and (**B**) six weeks post-injection with brimonidine–Laponite intravitreal formulation. Abbreviations: IF: intravitreal formulation; VIT: vitreous; RPE: retinal pigment epithelium; ILM: inner limiting membrane.

**Figure 2 pharmaceutics-13-00217-f002:**
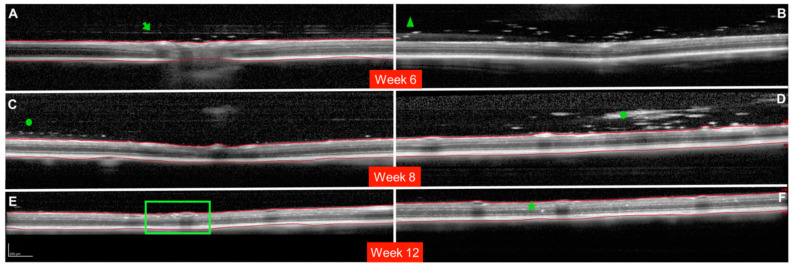
Milestones in the evolution of the brimonidine–Laponite intravitreal formulation (BRI/LAP IF) analysed using optical coherence tomography. (**A**): Cross-sectional image of the optic nerve showing hyper-reflective IF aggregates after crossing the posterior vitreous cortex (PVC). The green arrow points to the posterior vitreous cortex. Three hyper-reflective dots are found in the space between the PVC and the inner limiting membrane (ILM). (**B**): Hyper-reflective aggregates (green triangle) at the moment of crossing the posterior vitreous cortex. (**C**): Hyper-reflective aggregates (green circle) arranged one-by-one in a row. (**D**): Large BRI/LAP IF aggregate (green rhombus) in the vitreous humour. A light optical shadow can be observed (indicating potential perception of floaters) similar to the shadow that retinal vessels produce. (**E**,**F**): Hyper-reflective aggregates penetrating the retinal layers. Deposits in the inner nuclear layer or perivascular (green square). Deposits in the outer nuclear layer (green star). Red lines indicate the ILM and retinal pigment epithelium (RPE) boundaries. (**B**) Shows an example without boundaries (red lines) so as to permit measurement. (**A**,**B**) Show images obtained at 6 weeks of follow-up. (**C**,**D**) Show images obtained at 8 weeks of follow-up. (**E**,**F**) Show images obtained at 12 weeks of follow-up. Representative images are extracted from different animals.

**Figure 3 pharmaceutics-13-00217-f003:**
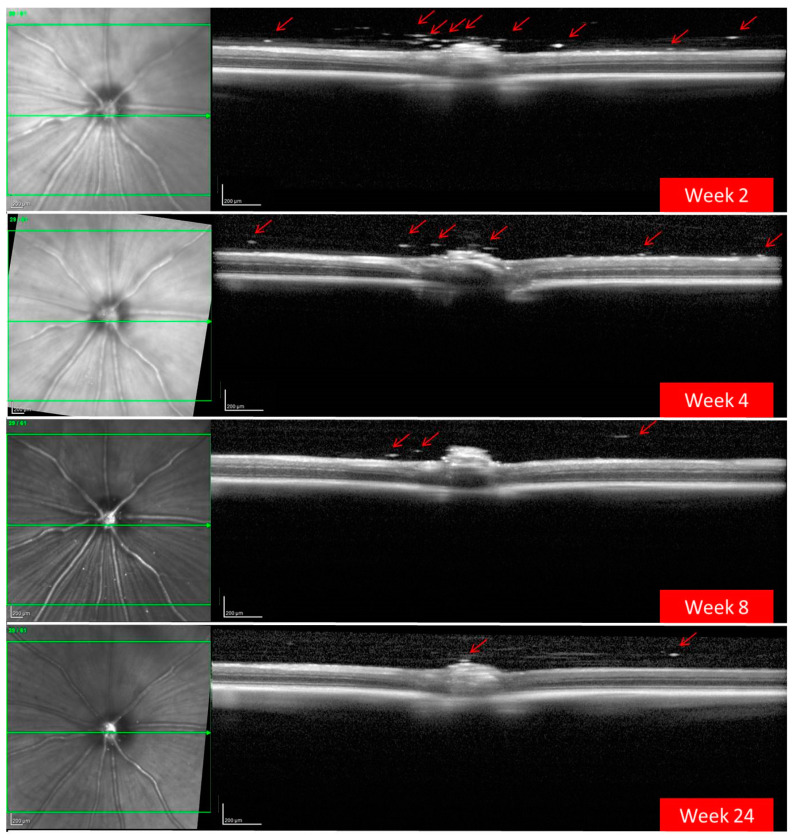
Progressive decrease in hyper-reflective aggregates (red arrows) of the brimonidine–Laponite intravitreal formulation (BRI/LAP IF) detected in the vitreous–retinal interface using optical coherence tomography over 24 weeks of follow-up.

**Figure 4 pharmaceutics-13-00217-f004:**
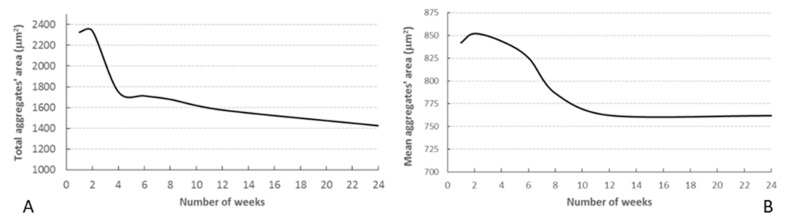
Temporal change in aggregate area. (**A**) Total area of the aggregates located in the vitreous; (**B**) mean area of the aggregates.

**Figure 5 pharmaceutics-13-00217-f005:**
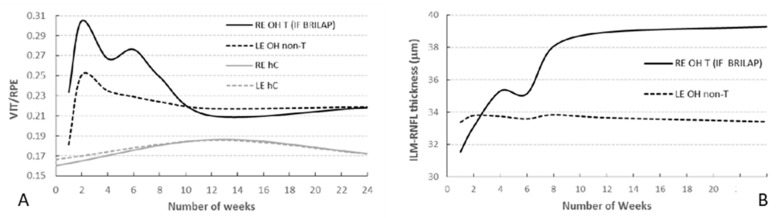
(**A**) Mean VIT/RPE relative intensity in rats with induced bilateral glaucoma (RE treated with brimonidine–Laponite) and healthy controls over 24 weeks of follow-up. (**B**) Segmentation of the ILM–RNFL in rats with induced bilateral glaucoma (RE treated with brimonidine–Laponite) over 24 weeks of follow-up. Abbreviations: RE: right eye; LE: left eye; OH: ocular hypertension; T: treated; BRI/LAP IF: Brimonidine–Laponite intravitreal formulation; non-T: non-treated; hC: healthy control; ILM: inner limiting membrane; RNFL: retinal nerve fibre layer.

**Figure 6 pharmaceutics-13-00217-f006:**
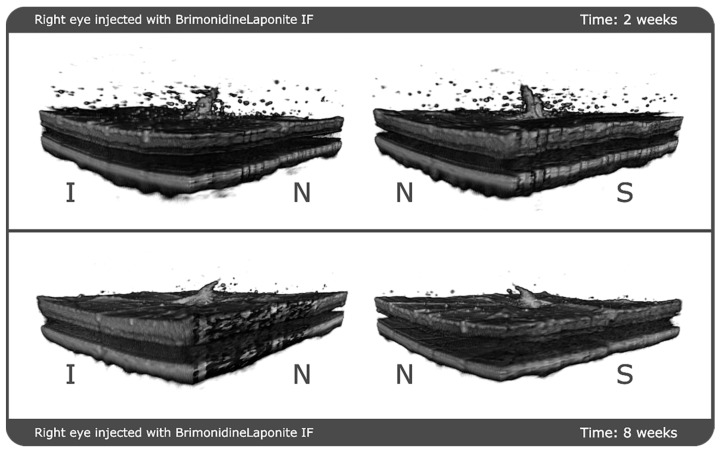
3-D reconstruction of the evolution of the aggregates at 2 weeks and 8 weeks of follow-up. The reconstruction is shown from two different perspectives at each point in time. Abbreviations: N: nasal; I: inferior; S: superior; IF: intravitreal formulation.

**Figure 7 pharmaceutics-13-00217-f007:**
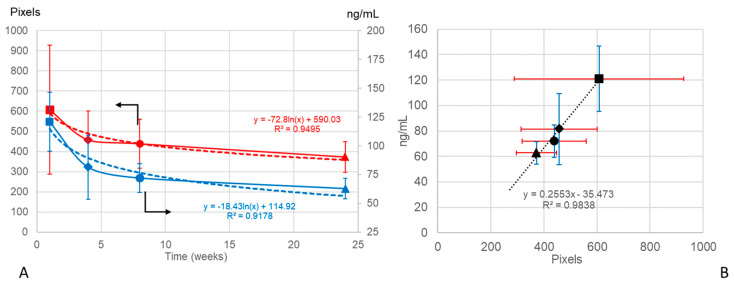
(**A**) Decreasing brimonidine level curves in rat eyes expressed in ng/mL (mean ± standard deviation; *n* = 3 eyes in each study time) (data from [9] (CC BY 4.0 license)) (in blue) and total aggregate area in the rat eye vitreous expressed in pixels (mean ± standard deviation; *n* = 9 eyes at week 1, *n* = 21 eyes at week 4, *n* = 8 eyes at week 8 and *n* = 5 eyes at week 24), obtained using optical coherence tomography (in red) over 24 weeks of follow-up. Logarithmic curves in dashes. (**B**) Positive linear correlation between drug levels and total aggregate area. Data are expressed as means ± standard deviation; optical coherence tomography (OCT) data in red; brimonidine data in blue; ■: 1 week; ◆: 4 weeks; ●: 8 weeks; ▲: 24 weeks.

**Figure 8 pharmaceutics-13-00217-f008:**
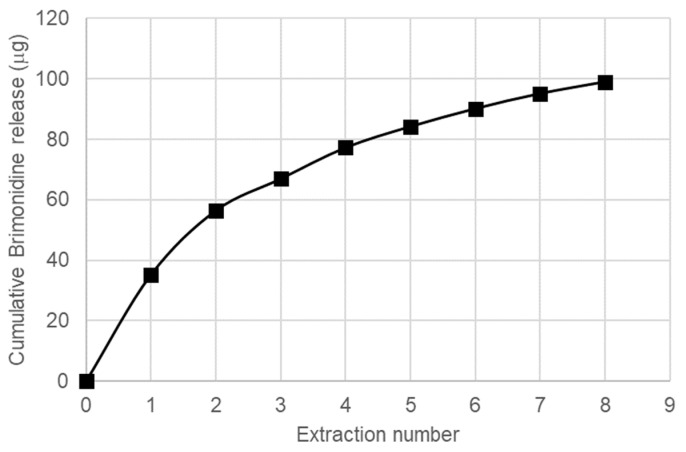
Cumulative release of BRI from BRI/LAP to the model of vitreous humour.

**Table 1 pharmaceutics-13-00217-t001:** Effect of BRI/LAP IF on intraocular pressure.

TIME	OHT (>20 mmHg) EYES (in %)		Intraocular Pressure (X ± sd)		
	Non-Treated	Treated	Non-Treated	Treated	*p*
BASELINE	0	0	9.12 ± 1.48	8.86 ± 1.69	0.476
2 w	88	4.8	23.34 ± 3.53	14.96 ± 4.16	<0.001
4 w (1 m)	91.7	28.1	25.26 ± 3.69	17.36 ± 4.10	<0.001
6 w	100	58.8	27.22 ± 3.15	20.64 ± 5.04	<0.001
8 w (2 m)	95	43.8	28.93 ± 7.11	19.85 ± 4.51	<0.001
12 w (3 m)	36.8	50	19.10 ± 3.07	19.70 ± 2.39	0.423
16 w (4 m)	58.3	80	20.05 ± 4.35	21.79 ± 1.42	0.364
20 w (5 m)	28.6	60	17.38 ± 2.87	21.19 ± 5.49	0.166
24 w (6 m)	71.4	60	23.66 ± 5.45	23.26 ± 4.82	0.684

Abbreviations: BRI/LAP IF. Brimonidine/laponite intravitreal formulation; OHT: ocular hypertension; w. week; m: month; *p* < 0.05: statistical significance; %: percentage; X ± sd: media ± standard deviation. Data from [9] (CC BY 4.0 license).

## Data Availability

The data presented in this study are available on request from the corresponding author.

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
