# Peer review of "Monitoring New Long-Lasting Intravitreal Formulation for Glaucoma with Vitreous Images Using Optical Coherence Tomography"

_pharmaceutics, 2021, doi:10.3390/pharmaceutics13020217_

Round 1
Reviewer 1 Report
the paper presents a new method to evaluate the presence of paricles of drug delivery system in a mose model of glaucoma. the approach may be interesting but using the OCT system is not really a complete examination of vitreoretinal interface, so that a great number of particles might be unevaluated due to the specificity of the technique that does not reach peripheral areas and usuakky are limited to the posterior pole, particularly in small eyes animals. it is hard to evaluate the progressive reabsorbtion of drug in terms of hyperintensity or in terms of size because limitation of the technique. might be interesting evaluate the effects of the drug on intraocular pressure
Optical Coherence Tomography it is a very well known technique for the analysys of the retna either in Human eyes or in experimental models, obviously using this device in very small animals requires compensative lens to reach a good optical quality. the paper even it is interesting has major bugs and particularly in my opinion could be very interesting to know if the reduction time correlated of particules inside the eye as demonstrated by OCT can control intraocular pressure and for how much time.
moreover the device is really affordable for the studi of central retinal area but does not reach more peripheral retina and there are aberrations that may be more important in evaluating the size of particles.
these factors might be addressed before reconsider the paper for pubblication
Reviewer 2 Report
The author showed that a qualitative and quantitative method of using OCT to analyze the signal generated in the vitreous by the BRI/LAP IF for glaucoma treatment. This qualitative and quantitative method is interesting, and manuscript is well written. However, the experimental number in most important data of Fig. 7 is few, and it has not been examined whether it can be used for other model animals. I think that it need to add the result using rabbit and/or other animals. Our comments are followings.
1) Figure 7: Author detected the particles of laponite in the vitreous humor, and showed that the changes in particles reflected the brimonidine levels in the Fig. 7. Although, the high correlation is observed (R value is high), the data number is only 4 point. Authors must increase measurement points or show all measured data in the Fig. 7B. In addition, the S.D. should be also add in the Fig. 7A and 7B.
2) In this study, author used Long-Evans rat. It is necessary to clarify whether these qualitative and quantitative method of using OCT are applicable to other models such as rabbits, since eye of rat is small, and the lens structure is also different from humans. Please add the result using rabbit and/or other animals with an organizational structure similar to humans.
Reviewer 3 Report
A very interesting paper where OCT has been used to follow the effect of brimonidine-Laponite. Some concerns I have:
- More information on the preparation, purification and characterisation of the brimonidine-Laponite (BRI/LAP) complex would be useful - this may have been described in a previous publication, but an overview is needed here.
- Is there any in vitro data to demonstrate the release kinetics of the brimonidine from the Laponite?
- Can you be more precise about where the dose was administered?
- In addition to the untreated conrols, i feel a further control is needed, namely, injection of the blank Laponite. Whereas I believe the changes seen are due to the drug, we cannot be certain based on the data shown.
Reviewer 4 Report
Well written manuscript overall. A couple of minor comments to address:
Does the hyperreflectivity of the formulation affect the vision of an individual? This (and any considerations around it) should be mentioned at some point in the manuscript.
Line 111 – please state that it is 1536 x 496 pixels per image.
2.3. – Was a calibration curve or other method optimised to yield a quantitative value for the aggregates? Was a minimum limit defined for the aggregate size?
Line 206 – It was not clear why this effect could be produced by the induction of glaucoma. Could the authors please elaborate?
Figure 5A – Is this figure indicating that the signal intensity could be confounded by the presence of glaucoma? How did the authors plan to utilise this data set for quantitation?
The points in lines 374-377 should be better elaborated on in the methods. This technique looked specifically at the vitreo-retinal interface and not the entire vitreous. While it was still correlated with drug concentrations from the previous study, the article was misleading to the reviewer making me initially anticipate a full vitreous humour characterisation of the particles.
It also appears that this technique is only suitable for hyper-reflective particles. Please be clear in the manuscript that the technique is tailored to these specific particles and wouldn’t be able to quantify any old therapeutic injected into the vitreous humour.
Round 2
Reviewer 2 Report
The authors answered all questions, and mentioned further improvement points in this study. So, I recommend the publication of the manuscript.
Author Response
thank you for recommending our manuscript for publication.